# Integrating Fly Ash-Controlled Surface Morphology and Candle Grease Coating: Access to Highly Hydrophobic Poly (L-lactic Acid) Composite for Anti-Icing Application

**DOI:** 10.3390/nano13071230

**Published:** 2023-03-30

**Authors:** Zhiqiang Jiang, Bai Xue, Xiaoping Mai, Changmei Wu, Lingjun Zeng, Lan Xie, Qiang Zheng

**Affiliations:** 1Department of Polymer Materials and Engineering, College of Materials and Metallurgy, Guizhou University, Guiyang 550025, China; zqjiang0120@126.com (Z.J.); mm58272000@163.com (X.M.); wcm1093069705@163.com (C.W.); zlj2740386528@163.com (L.Z.); 2State Key Laboratory of Public Big Data, Guizhou University, Guiyang 550025, China; 3National Engineering Research Center for Compounding and Modification of Polymer Materials, National and Local Joint Engineering Research Center for Functional Polymer Membrane Materials and Membrane Processes, Guiyang 550014, China; 4Department of Polymer Science and Engineering, Zhejiang University, Hangzhou 310027, China; zhengqiang@zju.edu.cn

**Keywords:** fly ash, phase separation, hydrophobicity, candle grease, petal-like microsphere

## Abstract

New ways of recycling fly ash are of great significance for reducing the environmental pollution. In this work, biodegradable hydrophobic poly (L-lactic acid)/fly ash composites for anti-icing application were successfully fabricated via a facile solvent-volatilization-induced phase separation approach. A silane coupling agent of 3-(Trimethoxysilyl) propyl methacrylate was used to decorate a fly ash surface (FA@KH570) for strengthening the interface bonding between fly ash and poly (L-lactic acid). Moreover, FA@KH570 could obviously enhance the crystallinity of poly (L-lactic acid) (PLLA)/FA@KH570 composites, which accelerated the conversion from the liquid-liquid to the liquid-solid phase separation principle. Correspondingly, the controllable surface morphology from smooth to petal-like microspheres was attained simply by adjusting the FA@KH570 content. After coating nontoxic candle grease, the apparent contact angle of 5 wt% PLLA/FA@KH570 composite was significantly increased to an astonishing 151.2°, which endowed the composite with excellent anti-icing property. This strategy paves the way for recycling waste fly ash and manufacturing hydrophobic poly (L-lactic acid) composite for potential application as an anti-icing material for refrigerator interior walls.

## 1. Introduction

As we all know, poly (L-lactic acid) (PLLA) is a linear aliphatic polyester with excellent biodegradability [1], solvent resistance [2], and biocompatibility [3], and good mechanical properties [4], abundant resource [5], and wide application. Thereby, PLLA can be used as a potential biomass matrix material for anti-icing [6,7], tissue repair [8], organ replacement [9], oil-water separation [10], and so on. There are various methods for preparing hydrophobic PLLA materials, including the template method [11], electroplating method [12], electrospinning method [13], phase separation method [14], sol-gel method [15], chemical grafting method [16], etc. Evidently, the electrospinning process and sol-gel methods are cumbersome and energy-intensive, but phase separation is considered a simple and feasible method for efficient preparation of hydrophobic porous materials [17,18].

Based on the theoretical equation of Huggins: (1)$$\Delta G=RT\left(n_{s}\ln\phi_{s}{+n}_{\mathrm{PLA}}\ln\phi_{\mathrm{PLA}}{+\chi n}_{s}\ln\phi_{\mathrm{PLA}}\right)$$

Refs [7,19], the phase separation effect usually occurred when the Gibbs free energy of a single-phase system was higher than that of the two-phase system, which could be brought about by altering the temperature and/or introducing a certain amount of nonsolvent [20]. Liu et al. reported a strong superhydrophobic inorganic-organic hybrid coating by combining fluorinated ethylene propylene (FEP) and aluminum dihydrogen phosphate (ADP) using a creative two-step phase separation approach, which had potential applications in many fields [21]. Wang et al. proposed a water-assisted thermal shock phase separation method for the first time to prepare a kind of PLA foam with special micro-nano structures and a large contact angle up to 151° [22]. However, the tedious TIPS process consumed a lot of energy. On the contrary, NIPS can be more easily achieved by adding some certain nonsolvent. Zhao et al. prepared petal-shaped PLA composite membranes with adjustable water droplet adhesion performance by the NIPS method [23]. Su et al. used a simple NIPS method to obtain environment-friendly super-hydrophobic three-dimensional stereo-PLA composites by adjusting the thickness and surface roughness [24].

It was easily found that the two major factors for the construction of hydrophobic surfaces were reducing surface energy and constructing micro-nano structures [25]. Natheless, the micro-nano structures constructed by phase separation were relatively isolated. To address this problem, several reports widely indicated that introducing nanoparticles and stacking them on the surface was a common strategy to obtain complex micro-nano structures with a certain roughness for the hydrophobic property [25,26]. Sun et al. introduced zinc oxide modified by KH570 into PLLA and constructed a rough surface with a high contact angle of 146° by NIPS method [27]. Zhang et al. prepared a PLA/γFe_2_O_3_ membrane by electrospinning, and the PLA/γFe_2_O_3_ membrane possessed high porosity [28]. Wu et al. obtained a superhydrophobic membrane for gas absorption by using hydrophobically modified SiO_2_ nanoparticles to induce PVDF precipitation [29]. Most of the nanoparticles used for hydrophobic modification were ceramic nanoparticles, which were relatively expensive and high-density. Therefore, choosing a cheap and recyclable particle to fabricate hydrophobic materials has become a hot research topic.

Fly ash (FA) mainly comes from the flue gas emitted by burning coal during the process of thermal power generation [30], metal smelting [31], heating for warmth, and so forth. FA usually exhibited a smooth surface like glass [32], with an average particle size of less than 20 microns [33], and contains a variety of active components such as unburned carbon and metal oxides (SiO_2_, Al_2_O_3_, and Fe_2_O_3_ [34,35]). According to previous researches and reports, FA had been effectively and deeply applied in the fields of building materials [36], metallurgy [37], agriculture [38], and chemical industry [39] at home and abroad. However, less than half of FA was recycled as raw material; the residue was simply landfilled or piled up. The long-term stacking of FA not only pollutes soil and water but also does harm to human health [40]. Hence, the recovery and reuse of FA had become an urgent problem to be solved.

New routes of recycling FA were of significance for producing high-value materials and reducing environmental pollution. It is feasible to fabricate functional composites by incorporating FA into polymers [41]. Nonetheless, FA could not be well distributed into polymer matrix due to poor compatibility, which would lead to the deterioration of properties, especially mechanical properties. Thus, FA was normally modified to improve the compatibility and/or to achieve a new function; modification methods included physical modification such as mechanical modification [42], microwave modification [43], high-temperature roasting modification [44], and so on, and chemical modification such as acid modification [45], alkali modification [46], salt modification [47], organic reagent modification [48,49], and so forth. The common silane coupling agent KH570 could be facilely applied to chemically modify FA and graft long-chain alkanes on the surface [50]. As expected, the KH570-modified FA (FA@KH570) particles exhibited excellent compatibility with PLLA.

To the best of our knowledge, the introduction of FA into a PLLA matrix to prepare hydrophobic composites has been rarely reported. Herein, a facile solvent-volatilization induced phase separation approach is proposed to obtain hydrophobic PLLA/FA@KH570 composites. The surface hierarchical microstructures converted from smooth to petal-like microspheres merely by regulating the FA@KH570 loading. In addition, candle grease was melt-coated on the PLLA/FA@KH570 composite surface to further improve the hydrophobic property. PLLA/FA@KH570 composite (5 wt%) exhibited the maximum contact angle of 151.2°, resulting in a fascinating anti-icing property. This investigation opened up a new avenue for recycling waste FA to fabricate highly hydrophobic PLLA composites with a favorable anti-icing property.

## 2. Materials and Experimental Methods

### 2.1. Materials

The poly (L-lactic acid) (PLLA) granules (commercial mark: 4032D) with $\overset{-}{Mw}$ of 11.9 × 10^4^ g/mol and $\overset{-}{Mn}$ of 6.6 × 10^4^ g/mol were procured from Nature Works (Blair, NE, USA). Fly ash (grade: first-class fly ash, diameter: 8–12 µm, purity: 99%, density: 2.1 × 10^3^ kg/m^3^, and melting temperature: 1300 °C) was supplied by Jinchuan Stone Factory (Jining, China). Anhydrous ethanol and dichloromethane were acquired from Tianjin Fuyu Fine Chemical Co., Ltd. (Jianjin China). Silane coupling agent 3-(Trimethoxysilyl) propyl methacrylate (KH570) (density: ≈1 g/cm^3^) was provided by Nanjing You Pu Chemical Co., Ltd. (Nanjing, China). Candles were obtained from supermarkets. Distilled water was produced in the laboratory.

### 2.2. Modification of FA

FA containing a variety of active functional groups was conducive to modification. Typically, 2 g of FA was firstly distributed in a solution of 45 mL ethanol and 5 mL deionized water, and then transferred into a three-neck flask. Subsequently, 1 mL of KH570 was added into the above dispersion by a pipette under continuous mechanical stirring at 65 °C for 1 h. After that, the dispersion was subjected to ultrasonic treatment for 2 h in an ultrasonic cleaning machine. Finally, the modified FA powders (FA@KH570) were collected by filtration and transferred to an oven for drying at 45 °C for 24 h. After weighing, the powder weight was 1.57 g and the yield was 52.3%.

### 2.3. Preparation of PLLA/FA@KH570 Composites

As can be seen from Figure 1, PLLA/FA@KH570 composites were prepared via the phase separation induced by solvent evaporation. Firstly, a predesigned quantity of FA@KH570 was added to 20 mL of dichloromethane solution. After ultrasonically dispersing for 30 min, 1 g PLLA was mixed with the FA dispersion under mechanical stirring for 2 h. Then, 5 mL of the mixed solution was poured into a petri dish (diameter: 6 cm). A quantity of 2 mL non-solvent (anhydrous ethanol) was gradually dripped to induce phase separation. Finally, a series of PLLA/FA@KH570 composites were successfully prepared, including 1, 3, 5, and 7 wt%-PLLA/FA@KH570. For further enhancing the hydrophobicity, a small piece of candle (0.05 g) purchased from the supermarket was scraped onto the surface of the prepared PLLA/FA@KH570 composites, and then heated at 60 °C for 20 min to achieve an even distribution of candle grease.

### 2.4. Characterizations

FA and FA@KH570 were tested by Fourier-transform infrared spectroscopy (FTIR, IS50, Thermo Nicolet, St. Bend, OR, USA) to analyze the characteristic groups in the wavelength range from 4000 to 500 cm^−1^. The surface microstructures of FA and PLLA/FA composites were observed on a scanning electron microscope (SEM, model SU8010, HITACHI, Japan) at an acceleration voltage of 10 kV. The surface of the sample was sprayed with gold to increase the electrical conductivity before observation. The melting/crystallization behaviors of pure PLLA and PLLA/FA composites were explored using a differential scanning calorimeter (DSC, Model Q2000, manufacturer, TA Instruments, Newcastle, DE, USA) under a nitrogen atmosphere. The samples were heated from 10 °C to 200 °C at an ascending rate of 10 °C/min. To determine the crystalline structures, the specimens were scanned from 5° to 80° at a rate of 10°/min using an X-ray diffractometer (XRD, model, XPertPRO manufacturer, PANalytical, Almelo, The Netherlands) at ambient temperature. A contact angle measuring instrument (model JC2000D1 manufacturer, Shanghai Zhong Chen Digital Technology Equipment Co., Ltd., Shanghai, China) was employed to measure the apparent contact angle and rolling angle of the sample. Distilled water was injected into the provided needle tube, and the droplet size was determined by the cross-sectional area of the needle. Five points of each sample were selected for contact angle test, and the average value was taken. The abrasion stability of the contact angle for PLLA/FA@KH570 composites was evaluated by sandpaper abrasion measurement. The composite films were pressed on sandpaper (800 mesh) under a 200 g loading and slipped for 10 cm with an external force. Ultimately, the water contact angle of the abrased films was measured again. The anti-icing properties were characterized by simply measuring the weight evolution and delay time during the icing process. To simulate the frigid natural environment, the samples were placed in a refrigerator (the temperature was accurately retained at −18 °C and the humidity was adapted to 75% by a humidifier). The weight of the samples was obtained on an analytical balance at fixed intervals. In addition, the delay time was also tested on the above conditions. A deionized water droplet of 10 µL was dropped on the sample surface through a pipette. The icing point was defined as the moment that the water droplet changed from translucent to opacified. The average delay time of three tests was taken for each sample.

## 3. Results and Discussion

### 3.1. Dispersibility and Morphology of FA@KH570

For the sake of proving the dispersibility of the FA@KH570 microparticles, unmodified FA and FA@KH570 microparticles were dispersed in dichloromethane solution and deionized water under ultrasonication, respectively (Figure 2). After standing for 2 h, it was clearly seen that FA microparticles were in a turbid state in the water (Figure 2a), which was the proof that FA microparticles were dispersed well in the water. However, most of FA microparticles were deposited at the bottom in dichloromethane due to the poor dispersibility and high density (Figure 2a_1_). FA@KH570 microparticles showed the opposite state: FA@KH570 floated on the surface in aqueous dispersion due to its hydrophobicity (Figure 2b); FA@KH570 was uniformly dispersed in dichloromethane, indicating the excellent compatibility of FA@KH570 with dichloromethane (Figure 2b_1_). From the optical image in Figure 2c, the water droplet would directly penetrate into FA microparticles, further illustrating the hydrophilicity of FA. On the contrary, FA@KH570 repelled the water droplet, owing to the grafting of long-chain alkanes (Figure 2c_1_).

In addition, FTIR spectroscopy was applied to characterize the chemical structures of FA@KH570 and pure FA (Figure 3a). It could be found that both FA@KH570 and pure FA had a strong peak at 3462 cm^−1^, which was ascribed to the stretching vibration of -OH. The strong band at 1077 cm^−1^ was indicative of the asymmetric stretching vibration of Si/Al-O. It was worth noting that a new band was observed at 1725 cm^−1^, belonging to the stretching absorption peak of C=O. The macromolecules with hydroxyl in the hydrolyzed products of KH570 could form covalent bonds by chemical bonding with –OH on the surface of FA [51]. The stretching absorption peak of C=O existing in FA@KH570 proved that KH570 was successfully grafted on the FA surface. Meanwhile, the XRD tests were applied to explore the crystalline structures of FA and FA@KH570 (Figure 3b and Appendix A). It was easy to find that FA exhibited the diffraction peaks located at 18.2° and 26.7°, corresponding to the crystal phase quartz. Therein, the carbon content was low and the characteristic peaks were covered with quartz, which was imperceptible in the pattern. The diffraction peak observed at 40.1° was assigned to the crystal of mullite. Moreover, the crystallization peaks of hematite were prevailingly focused on 33.3°, and the characteristic peak of alumina appeared at 35.4°. It was noteworthy that FA@KH570 illustrated similar diffraction peaks to FA, demonstrating that KH570-modification had little effect on the crystalline structures of FA. As can be seen from the morphology of FA and FA@KH570 (Figure 3c,d), the modification of KH570 also scarcely changed the morphology and microstructures of FA.

### 3.2. Morphology of PLLA/FA Composites

As can be seen from Figure 4a,a_1_ and Appendix A, the surface of pure PLLA was relatively smooth with a small quantity of microspheres (diameter: <2 µm). The morphology of PLLA/FA@KH570 composites exhibited a sequence of changes with the increase in the FA@KH570 fraction from 1 wt% to 7 wt% (Figure 4b–e and Appendix A). In comparison with pure PLLA, 1 wt% PLLA/FA@KH570 possessed more and larger microspheres (diameter ≈ 2.5 µm) on the surface, due to the effective nucleation of FA@KH570 (Figure 4b,b_1_). For 3 wt% PLLA/FA@KH570, the number of microspheres was continuously increased with the incremental FA@KH570 loading. Strikingly, petal-like microspheres with multilayer nano-wrinkles were observed on the 5 wt% PLLA/FA@KH570 composite surface (Figure 4c,c_1_). The production of petal-like microspheres was associated with the conduct of two diverse phase separation processes, that is to say, quick liquid-solid phase separation stemming from heterogeneous nucleation of PLLA and FA@KH570, generating many micro-domains where tardy liquid-liquid phase separation was next in progress [52,53]. However, when the content of FA@KH570 was increased to 7 wt%, the number of microspheres began to decline, on account of the severe aggregation of FA@KH570 (Figure 4e,e_1_). In short, the petal-like microspheres attached on the surface provided multiscale micro-nano structures, which favors hydrophobicity.

The particular mechanism for the construction of hierarchical microstructures via the solvent-volatilization-induced phase separation approach is vividly modeled in Figure 5. The phase separation would take place when the Gibbs free energy of the PLLA solution violently fluctuated. In this investigation, the solution would be stratified as a result of the density difference between the solvent (dichloromethane, 1.33 g/cm^3^) and nonsolvent (ethanol, 0.79 g/cm^3^). Since dichloromethane volatilized much more quickly than ethanol, concentration fluctuation appears at the solvent/nonsolvent interface, resulting from PLLA chain agglomeration induced by the decreasing free energy of this system. Hence, the molecular chains could undergo interphase migration along the concentration gradient direction and trigger the formation of polymer-lean and polymer-rich phases through the diffusion-induced separation. Thanks to the superior volatility of dichloromethane, phase inversion took place while the volume of the polymer-lean phase was elevated to a certain extent, which was accompanied by the transformation of the isolated polymer-lean phase into the continuous phase. The 3D-network polymer-rich phase would be compressed into big slices during phase inversion procedure, as a consequence of the increasing concentration and viscosity. Thereby, the fish-scale skeletons were constructed via the liquid-liquid phase separation. The imprint of polymer-lean phase assembled the pores throughout the 3D connected networks. 

In addition, a small number of homogeneous crystal (HC) nuclei in the polymer-rich phase were already formed by the orderly packing of the closely adjacent pre-ordered polymer chains. These nuclei would grow into microspheres by the stacking of PLLA chains along the phase interface, which was attributed to the increased viscosity and restrained mobility of PLLA chains. Regarding PLLA/FA@KH570 composites, FA@KH570 could effectively improve the heterogeneous nucleation of PLLA and thus generate many more crystal nuclei. After crystal growth, more and larger microspheres were assembled on the composite surface. Notably, at the FA@KH570 content of 7 wt%, the serious aggregation of FA@KH570 largely retarded the movement of PLLA chains and the growth of crystals. Thus, the microspheres on the surface of 7 wt% PLLA/PLLA/FA@KH570 were greatly decreased in number.

### 3.3. Crystallization Behaviors of PLLA/FA@KH570 Composites

The crystallization properties of the specimens were explored by DSC and XRD systems (Figure 6). The degree of crystallization of PLLA composites could be obtained by melting enthalpy divided by the theoretical enthalpy value for entire crystallization (93.1 J/g) [27,54]. It could be seen from the DSC heating curves of PLLA composites (Figure 6a) that the melting temperature (T_m_) of PLLA/FA@KH570 composites exhibited little variation (166.5–167.5 °C). From the glass transition temperature (T_g_) point of view, T_g_ was slightly increased from 55.9 °C for pure PLLA to 60.3 °C for 7 wt% PLLA/FA@KH570, which was mainly owing to the decreasing movement of PLLA chains with the adhesion of FA@KH570. Moreover, the crystallinity of pure PLLA was only 55.1%. The crystallinity of PLLA/FA@KH570 composites was improved from 55.5%, 68.9% to 78.2% with the incremental FA@KH570 content from 1 wt%, to 3 wt%, to 5 wt%. The underlying mechanism was that FA@KH570 could play a role as an excellent heterogeneous nucleating agent to accelerate the crystallization of PLLA composites [55]. Nevertheless, 7 wt% PLLA/FA@KH570 possessed a significantly reduced crystallinity of 65.7%, on account of the agglomeration of FA@KH570, largely reducing the movement of PLLA chains and limiting the growth of crystals.

As is well known, the crystal form of PLLA is dependent on the crystallization conditions, and there are generally the four crystal forms of α, α’, β, and γ. According to XRD image of PLLA composites (Figure 6b), the crystallization peaks were located at 2θ angles of 12.5°, 14.8°, 16.7°, 18.9°, and 22.4°, corresponding to (004)/(103), (010), (200)/(110), (014)/(203), and (015) crystal planes, respectively. Noticeably, the corresponding 2θ shifted to a slightly lower angle with the incremental FA@KH570 loading. This phenomenon could be explained by the fact that the increased FA@KH570 particles would lead to the larger unit cell parameters and interplanar spacing. In addition, the crystallization peak intensity was enhanced with the increase of FA@KH570 loading from 1 wt% to 5 wt%, and then decreased at a FA@KH570 loading of 7 wt%, in good agreement with DSC analyses. All in all, a small amount of FA@KH570 had little impact on the crystalline structures of PLLA.

### 3.4. Surface Wettability of PLLA/FA@KH570 Composites

The surficial wettability of composite materials heavily depended on the geometrical structure and surface energy. Nevertheless, a large quantity of polar ester groups was located in PLLA chains, leading to unsatisfactory hydrophobicity. Therefore, constructing proper surface microstructures with high roughness and reducing surface energy were keys to enable PLLA materials with hydrophobic properties. In this study, the rough surface of PLLA composites with crystalline microspheres and FA@KH570 particles was successfully assembled via a straightforward solvent-volatilization-induced phase separation method. Afterward, the PLLA/FA@KH570 composites were coated with candle grease by a simple melt-coating method to reduce the surface energy, finally exhibiting surface super-hydrophobicity [56].

The contact angle tests were carried out to explore the surface wetting property of the PLLA/FA@KH570 composites. As displayed in Figure 7a, the apparent contact angle of the PLLA/FA@KH570 composites was raised from 124.2° to 144.7° with an increase in the FA@KH570 content from 0 to 5 wt%. Surface wettability evolution was primarily dominated by the surface roughness of the PLLA/FA@KH570 composites. A gas cushion was built under the water droplet by a mass of entrapped air in the rough PLLA composite surface, which made the solid-liquid contact surface transform into a three-phase surface consisting of air, water, and solid. Thus, spherical droplets appeared on the PLLA/FA@KH570 composite surface, exhibiting greatly improved hydrophobicity. At the incremental FA@KH570 loading of 7 wt%, the apparent contact angle was exceptionally reduced to 131.6°. The underlying principle was that the surface roughness decreased, accompanied by a reduced number of microspheres on the 7 wt% PLLA/FA@KH570 surface, which had been confirmed by the SEM results. 

In order to achieve hydrophobic PLLA/FA@KH570 composites, commercially available and nontoxic candle grease, as a common low-surface energy substance, was coated on the PLLA/FA@KH570 surface by a facile melt-coating approach to reduce the surface energy. The apparent contact angle of PLLA/FA@KH570 composites after candle grease coating is depicted in detail in Figure 7b and Appendix A. It was obvious that the contact angle of PLLA/FA@KH570 composites was highly enhanced after candle grease coating at the same FA@KH570 loading fraction. For instance, the contact angle of 5 wt% PLLA/FA@KH570 composite after candle grease coating was elevated to the maximum value of 151.2°, compared with that of 5 wt% PLLA/FA@KH570 composite without candle grease coating (144.7°). PLLA/FA@KH570 composites after candle grease coating could show astonishing hydrophobicity. As shown in Figure 7c, the apparent contact angle of PLLA/FA@KH570 composites after sandpaper abrasion was slightly decreased. For example, the contact angle of 5 wt% PLLA/FA@KH570 was moderately reduced from 151° to 145° after abrasion, which still exhibited great hydrophobicity. The results indicated the excellent abrasion stability of PLLA/FA@KH570 composites.

According to the adhesion theory, rose petal-like surfaces were not only hydrophobic but also somewhat adherent. As could be seen from Figure 4, the surface of the 5 wt% PLLA/FA@KH570 composite had a petal-like structure. Correspondingly, we conducted a rolling angle test and the results were as expected (Appendix A). As shown in Appendix A, the advancing and receding contact angles of 5 wt% PLLA/FA@KH570 at a tilt angle of 60° are 155.4° and 108.2°, respectively. Thus, the contact angle hysteresis up to 47.2° indicates the excellent adhesion of the water droplet on the composite surface, in spite of the high static contact angle of 151°.

### 3.5. Anti-Icing Property

The highly hydrophobic PLLA/FA@KH570 composites opened up a promising application in the anti-icing field. To estimate the anti-icing properties, the weight increase (φ) and delay time during icing process (−18 °C and 75% RH) were measured in detail (Figure 8). The weight increase (φ) was calculated according to the following equation:(2)$$\varphi =\frac{W_{i}-W_{0}}{W_{0}}\times 100\%$$
where *W_i_* is the weight after icing at designed intervals and *W*_0_ is the primal weight of the specimen.

After an icing time of 24 h, the weight of both pure PLLA and PLLA/FA@KH570 composite changed little. The weight evolution of the samples was recorded within 24 h. As seen from Figure 8a and Appendix A, the weight of pure PLLA was sharply increased from 0.299 g to 0.339 g at 24 h, exhibiting a weight increase up to 13.4%. However, 5 wt% PLLA/FA@KH570 composite indicated a much lower weight increase of only 5.3%. Moreover, the delay time of 5 wt% PLLA/FA@KH570 composite was significantly prolonged to 410 s, in comparison with that of pure PLLA (merely 159 s). The improved anti-icing property stemmed from the enhanced hydrophobicity of 5 wt% PLLA/FA@KH570 composite, of which the apparent contact angle appreciably reached to the maximum value of 151.2°. On the PLLA/FA@KH570 composite surface, the water droplet trapped a lot of air in the micro-nanostructures to elicit the formation of an air cushion, immensely decreasing the contact area between the PLLA composite surface and the water droplet, largely weakening the surface adhesion, and effectively lessening the heat transfer. Considering all the above analyses, one could make a conclusion that highly hydrophobic PLLA/FA@KH570 composites are a great prospect in anti-icing smart materials. Concretely, it could be applied to the outer walls of electronic devices to prevent the corrosion of parts by water.

## 4. Conclusions

In summary, reusing FA is crucial to reducing environment pollution and producing high-value materials. First, FA was modified by KH570 (FA@KH570) to reinforce the interfacial bonding between FA@KH570 and PLLA. Then, PLLA/FA@KH570 composites were successfully fabricated via a facile solvent-volatilization-induced phase separation approach. FA@KH570 could play a role as an excellent heterogeneous nucleating agent to accelerate the crystallization of PLLA composites. Correspondingly, the crystallinity of the PLLA/FA@KH570 composites was increased from 55.1 to 78.2% with the slightly incremental content of FA@KH570 from 0 to 5 wt%. Thus, the surface morphology of PLLA/FA@KH570 composites could transform from smooth to petal-like microsphere by merely modulating the FA@KH570 loading. In addition, the surface of the PLLA/FA@KH570 composites was melt-coated with commercial candle grease to reduce the surface energy. It was worth noting that 5 wt% PLLA/FA@KH570 composite exhibited a maximum contact angle of 151.2°, due to the increased surface roughness and the reduced surface energy, according to Cassie wetting theory. In addition, the contact angle hysteresis of 5 wt% PLLA/FA@KH570 composite was up to 47.2°, indicating the excellent adhesion of the water droplet on the rose petal surface. The water droplet trapped air to create an air cushion on the rough PLLA composite surface, contributing to the greatly reduced contact area of the water droplet with the PLLA composite surface. Predictably, 5 wt% PLLA/FA@KH570 composite displayed an excellent anti-icing property, and deviated from the minimum weight increase of 5.3% and the maximum delay time of 410 s during the icing process. The FA@KH570-controlled surface morphology strategy represents a significant step towards recycling waste FA and fabricating green hydrophobic PLLA composites, which has tremendous implications for anti-icing materials and environmental protection. The PLLA/FA@KH570 composite films have great potential as a promising material for refrigerator interior walls.

## Figures and Tables

**Figure 1 nanomaterials-13-01230-f001:**
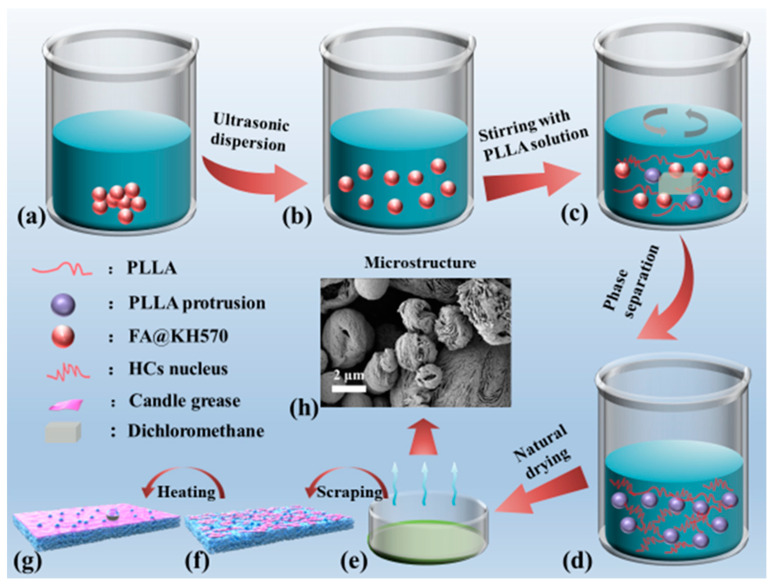
Schematic diagram of the preparation procedure of PLLA/FA@KH570 composites. (**a**) FA@KH570 in dichloromethane, (**b**) FA@KH570 dichloromethane dispersion, (**c**) PLLA/FA@KH570 dichloromethane solution, (**d**) PLLA/FA@KH570 gel, (**e**) PLLA/FA@KH570 composite, (**f**) candle-scraped PLLA/FA@KH570 composite, (**g**) heated candle-scraped PLLA/FA@KH570 composite, (**h**) Microstructure of PLLA/FA@KH570 composite.

**Figure 2 nanomaterials-13-01230-f002:**
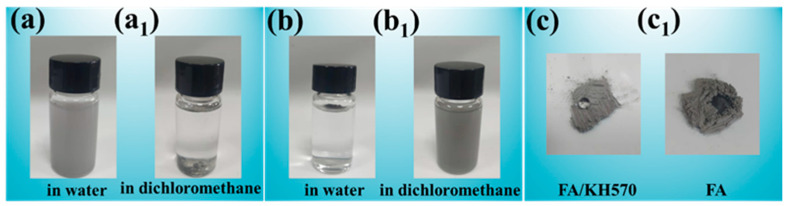
(**a**) and (**a_1_**) the photos of FA dispersion in water and dichloromethane, respectively. (**b**) and (**b_1_**) the photos of FA@KH570 dispersion in water and dichloromethane, respectively. (**c**) and (**c_1_**) the optical pictures of water droplet on the surface of FA@KH570 and FA, respectively.

**Figure 3 nanomaterials-13-01230-f003:**
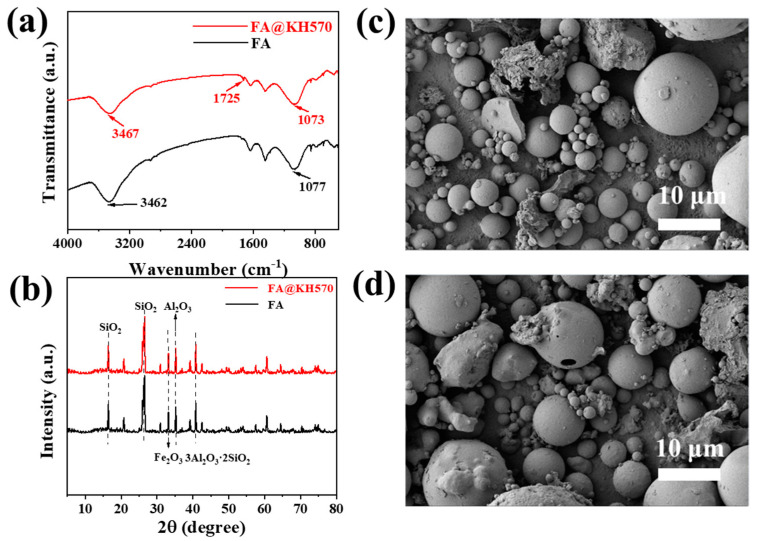
(**a**) FTIR of FA and FA@KH570. (**b**) XRD patterns of FA and FA@KH570. (**c**) and (**d**) SEM micrograms of FA and FA@KH570.

**Figure 4 nanomaterials-13-01230-f004:**
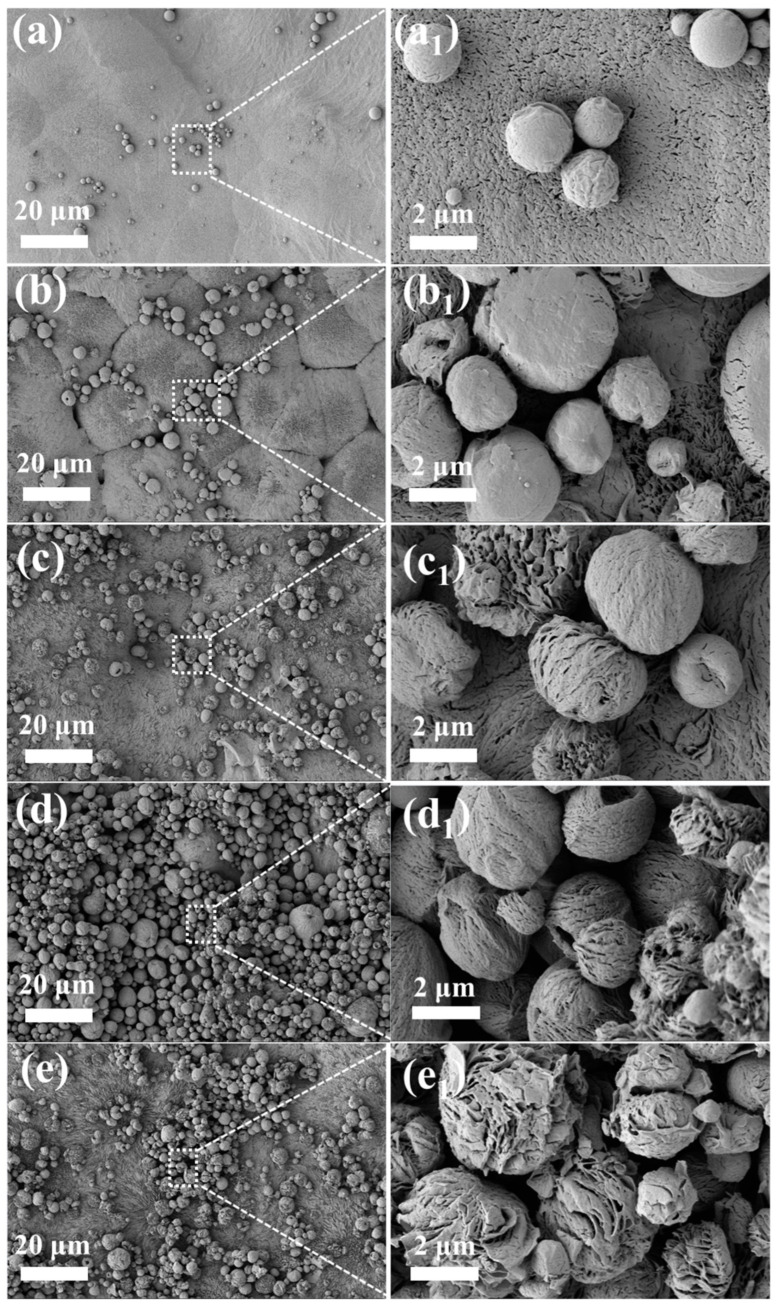
SEM images of (**a**,**a_1_**) pure PLLA, (**b**,**b_1_**) 1 wt% PLLA/FA@KH570, (**c**,**c_1_**) 3 wt% PLLA/FA@KH570, (**d**,**d_1_**) 5 wt% PLLA/FA@KH570, and (**e**,**e_1_**) 7 wt% PLLA/FA@KH570 composites.

**Figure 5 nanomaterials-13-01230-f005:**
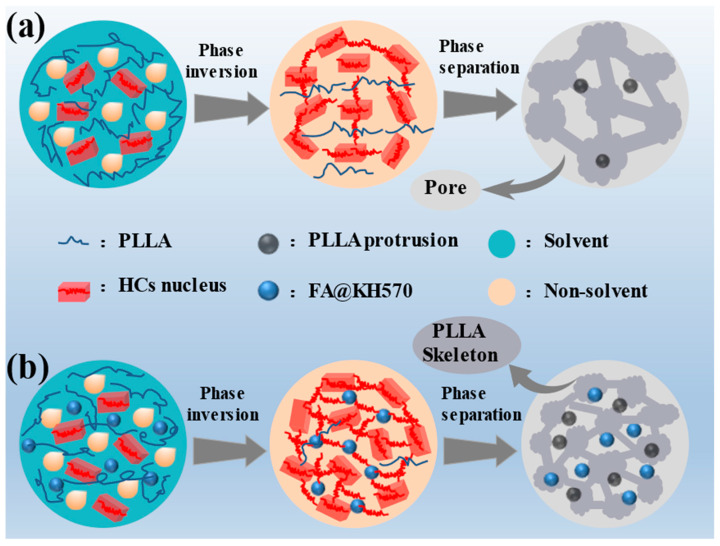
Mechanism diagram of the growth of hierarchical structures for (**a**) pure PLLA and (**b**) PLLA/FA@KH570 composites.

**Figure 6 nanomaterials-13-01230-f006:**
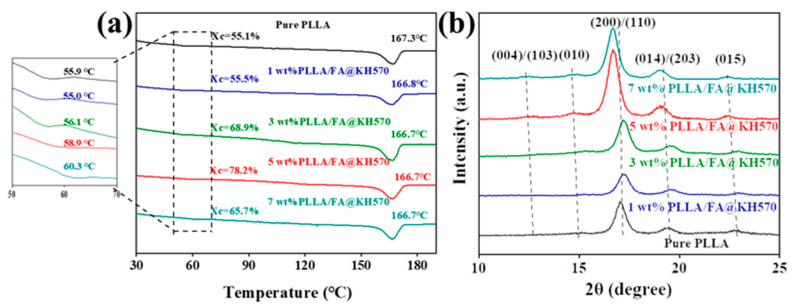
(**a**) DSC heating curves of pure PLLA and PLLA/FA@KH570 composites. (**b**) XRD patterns of pure PLLA and PLLA/FA@KH570 composites.

**Figure 7 nanomaterials-13-01230-f007:**
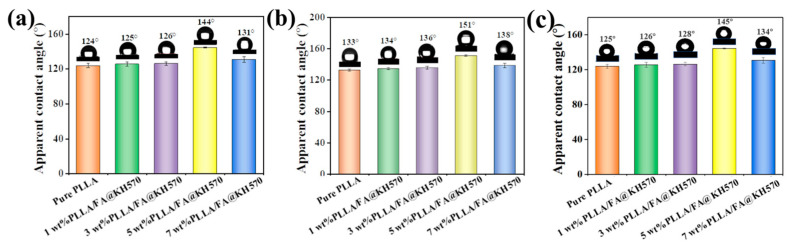
(**a**) Contact angles of neat PLLA and PLLA/FA@KH570 composites. (**b**) Contact angles of neat PLLA and PLLA/FA@KH570 composites with candle grease coating. (**c**) Contact angles of neat PLLA and PLLA/FA@KH570 composites after abrasion.

**Figure 8 nanomaterials-13-01230-f008:**
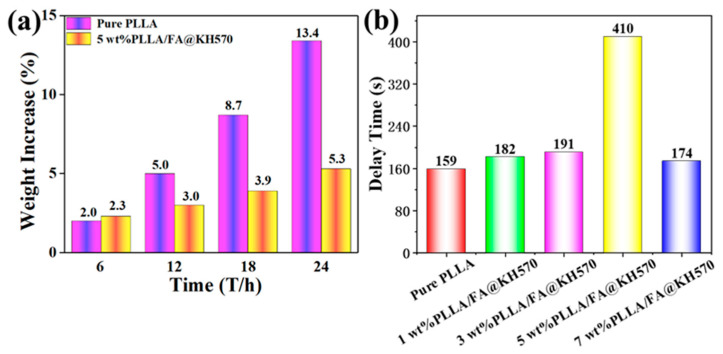
(**a**) Weight increase of pure PLLA and 5 wt% PLLA/FA@KH570 composite. (**b**) Delay times of neat PLLA and PLLA/FA@KH570 composites during the icing procedure.

## Data Availability

Data is contained within the article.

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
