# Peer review of "Integrating Fly Ash-Controlled Surface Morphology and Candle Grease Coating: Access to Highly Hydrophobic Poly (L-lactic Acid) Composite for Anti-Icing Application"

_nanomaterials, 2023, doi:10.3390/nano13071230_

Round 1
Reviewer 1 Report
In the reviewer's opinion, this is a solid study that contains a number of interesting aspects and results. Considering the title of the booklet and the laboratory "Big Data", I was a bit disappointed that the statistical data collection and evaluation e.g. with the contact angle analysis of HPDSA or similar methods like ADSA were not recognised. Nevertheless, I am convinced that the article can be published after some revisions.
Spezific comments
- When looking at the error bars in Figure 7, the first decimal place for the contact angles should be removed.
- Reduce the number of abbreviations in the abstract.
- The CA details are very rudimentary. You need to provide information on the type of contact (rose petal or no pinning). You will find some nice examples in the above mentioned references. The reviewer also strongly recommends taking a look at such references to see how a well-conducted CA experiment should be described and conducted. The easy way out is to give the reader information about the roll of angle and more details about the CA-experiment.
- I missed the information where the FA comes from.
- Line 187 “belonging to the stretching absorption peak of C=O. This proved that KH570 was successfully grafted on FA surface through condensation reaction of Si-OH.” This statement is as written I incomplete and simply wrong. You need to explain this in more detail. Please give the reader an estimate of the yield for the first synthesis step. I strongly suggest to optimize this step. You use 2 g FA and 1 g silane and see only a tiny signal. The tiny signal itself is not the problem, as you only need a tiny amount to change the surface (and the used IR and sample is not sensitive towards the surface), but the amount you use ?! Please search for "Silane Deposition via Gas-Phase Evaporation" and the cited publication to see the discrepancy.
- It is not comprehensible how the test surfaces for the CA measurements were produced. (area, solid content, substrate, film formation ?), e.g. the pure PLLA, …. (Following to Cytotoxicity Study of UV‐Laser‐Irradiated PLLA Surfaces Subjected to Bio‐Ceramisation: A New Way towards Implant Surface Modification, the PLLA surface has a CA in the range of 87° ?!) it is not clear if and how they are all performed using the solvent-volatilization induced phase separation approach and what is a HC nucleous: Scheme figure 5 have to be explained.
- XRD patterns of carbon are amorphous and very broad. Please carefully evaluate the patterns with hkl tables.
- The reviewer misses tables with the data on roughness, composition, Tm/g, weight increase, delay time and CA (and roll-off angle) of the different surfaces. So the “pretty hydrophilicity” cannot be seen
Some images in the supporting information can improve the quality.
Author Response
Response to the comments:
Dear Editors and Reviewers,
Thank you very much for your comments concerning our manuscript entitled “Integrating fly ash-controlled surface morphology and candle grease coating: access to highly hydrophobic poly (L-lactic acid) composite for anti-icing application” submitted to Nanomaterials. All the valuable comments have great help for our study and future work.
We have carefully read the comments and given detail answers as follows. We really appreciate and cherish this opportunity to answer these questions. A point-by-point revision to the reviewers’ criticisms has been made. Please find the final version with all corrections marked up using the “Track Changes”. We do hope our revised manuscript will meet your approval and be proceed further.
Sincerely yours,
Lan Xie
E-mail: mm.lanxie@gzu.edu.cn.
Response to Reviewer 1’s Comments:
Q1: When looking at the error bars in Figure 7, the first decimal place for the contact angles should be removed
A1: Thank you for your valuable suggestions which is of significance to improve our manuscript and work. The first decimal place for contact angles has been removed in Figure 7.
Figure 7. (a) Contact angles of neat PLLA and PLLA/FA@KH570 composites. (b) Contact angles of neat PLLA and PLLA/FA@KH570 composites with candle grease coating.
Q2: Reduce the number of abbreviations in the abstract.
A2: Thank you for your professional suggestions. The abbreviations in abstract have been reduced.
Abstract: New ways of recycling fly ash were of great significance for reducing the environmental pollution. In this work, biodegradable superhydrophobic poly (L-lactic acid)/fly ash composites for anti-icing application were successfully fabricated via a facile solvent-volatilization induced phase separation approach. Silane coupling agent of 3-(Trimethoxysilyl) propyl methacrylate was used to decorate fly ash surface (FA@KH570) for strengthening the interface bonding between fly ash and poly (L-lactic acid). Moreover, FA@KH570 could obviously enhance the crystallinity of poly (L-lactic acid) (PLLA)/FA@KH570 composites, which accelerated the conversion from liquid-liquid to liquid-solid phase separation principle. Correspondingly, the controllable surface morphology from smooth to petal-like microsphere was simply attained only by adjusting the FA@KH570 content. After coating nontoxic candle grease, the apparent contact angle of 5 wt% PLLA/FA@KH570 composite was significantly increased to astonishing 151.2°, which endowed the composite with excellent anti-icing property. This strategy paved the way to recycling waste fly ash and manufacturing superhydrophobic poly (L-lactic acid) composite for potential anti-icing application.
Q3: The CA details are very rudimentary. You need to provide information on the type of contact (rose petal or no pinning). You will find some nice examples in the above-mentioned references. The reviewer also strongly recommends taking a look at such references to see how a well-conducted CA experiment should be described and conducted. The easy way out is to give the reader information about the roll of angle and more details about the CA-experiment.
A3: Thank you for your valuable suggestions which is of significance to improve our manuscript and work.
Static contact angle testing was performed by contact angle measuring instrument and corresponding analysis software CAST3.0. The contact angle measuring instrument was equipped with a platform, needle tube and microscope. Place the test surface of the sample on a movable platform and dropwise add droplets from the needle tubing onto the surface of the composites. The image was developed by the analysis software and the contact angle was calculated. The average contact angle was obtained by measuring at any five points on the surface of PLLA composites. According to the adhesion theory, rose petal-like surfaces were not only hydrophobic but also somewhat adherent. As could be seen from Figure 4, the surface of the 5 wt% PLLA/FA@KH570 composite had a petal-like structure. Correspondingly, we conducted a rolling angle test and the results were as expected.
Figure S1. Static contact angles of pure PLLA (a) and 5 wt% PLLA/FA@KH570 composite (b), respectively.
Figure S5. Rolling angle photos of 5 wt% PLLA/FA@KH570 composite at the tilt angle of 30° (a) and 60° (b), respectively
As shown in Figure S5, the advancing and receding contact angles of 5 wt% PLLA/FA@KH570 at a tilt angle of 60° are 155.4° and 108.2°, respectively. Thus, the contact angle hysteresis up to 47.2° indicates the excellent adhesion of water droplet on the composite surface, in spite of the high static contact angle of 151°. The more detailed information is seen in Supporting Information (Figure S1-S6).
Q4: I missed the information where the FA comes from.
A4: Sorry, it’s our mistake to miss the information of FA.
Fly ash (grade: first-class fly ash, diameter: 8-12 µm, purity: 99%, density: 2.1×103 kg/m3, and melting temperature: 1300 °C) was supplied by Jinchuan Stone Factory (China).
Q5: Line 187 “belonging to the stretching absorption peak of C=O. This proved that KH570 was successfully grafted on FA surface through condensation reaction of Si-OH.” This statement is incomplete and simply wrong. You need to explain this in more detail. Please give the reader an estimate of the yield for the first synthesis step. I strongly suggest to optimize this step. You use 2 g FA and 1 g silane and see only a tiny signal. The tiny signal itself is not the problem, as you only need a tiny amount to change the surface (and the used IR and sample is not sensitive towards the surface), but the amount you use? Please search for "Silane Deposition via Gas-Phase Evaporation" and the cited publication to see the discrepancy.
A5: Thank you for the professional comments.
The macromolecules with hydroxyl in the hydrolyzed products of KH570 could form covalent bonds by chemical bonding with -OH on the surface of FA [51]. Thereinto, the stretching absorption peak of C=O existing in FA@KH570 proved that KH570 was successfully grafted on FA surface.
[51] Munief, W. M.; Heib, F.; Hempel, F.; Lu, X.; Schwartz, M.; Pachauri, V.; Hempelmann, R.; Schmitt, M.; Ingebrandt, S. Silane Deposition via Gas-Phase Evaporation and High-Resolution Surface Characterization of the Ultrathin Siloxane Coatings. Langmuir. 2018, 34, 10217-10229.
In this manuscript, FA is modified in KH570 solution, which is not a precise experiment. Quantitative graft is not needed, it only needs to obtain the hydrophobicity. The similar hydrophobicity of FA@KH570 is guaranteed by just maintaining the same modifying conditions. In the modification, only a small amount of KH570 is grafted on FA, but a majority of KH570 is left in the solution. Thus, a tiny single of C=O is seen in the FTIR spectrum. After weighing, the weight of resultant FA@KH570 is even less than 2 g, due to the loss from adhesion on bottle wall and filter paper. The FA@KH570 is weighted as 1.57 g and the yield is 52.3%.
Q6: It is not comprehensible how the test surfaces for the CA measurements were produced. (area, solid content, substrate, film formation?), e.g. the pure PLLA, …. (Following to Cytotoxicity Study of UV‐Laser‐Irradiated PLLA Surfaces Subjected to Bio‐Ceramisation: A New Way towards Implant Surface Modification, the PLLA surface has a CA in the range of 87°!) it is not clear if and how they are all performed using the solvent-volatilization induced phase separation approach and what is a HC nucleus: Scheme figure 5 have to be explained.
A6: Thank you for your valuable suggestions which is of significance to improve our manuscript and work.
The test surface of pure PLLA and PLLA/FA@KH570 directly grows from the solution via phase separation method. The full name of HC nucleus is homogeneous crystal nucleus, which is formed by orderly packing of the closely adjacent pre-ordered PLLA chains. The long PLLA chain can spontaneously form HC nucleus in methylene chloride. After the addition of non-solvent (ethanol), the polymer phase is compressed due to the faster volatilization rate of methylene chloride, which is more conducive to the formation of HC crystals and crystallization of PLLA. Moreover, FA@KH570 as a heterogeneous nucleating agent can further promote PLLA crystallization during the phase separation. As the ethanol and methylene chloride eventually evaporate, the polymer-rich phase becomes the fish-scale skeleton and the polymer-lean phase becomes the pore. Both homogeneous and heterogeneous crystal nucleus would grow into microspheres by the stacking of PLLA chains along the phase interface.
Due to a large number of polar ester groups on PLLA, the contact angle on smooth surface of pure PLLA is 50-90° [56]. But the rough surface structure can be obtained by phase separation method, the contact angle of rough PLLA surface can be improved to some extent. The contact angle of pure PLLA surface can be adjusted in the range of 90-130° only by changing the phase separation conditions (PLLA concentration, precipitation temperature, and so on) [R1-R5].
[56] Szustakiewicz, K., Kryszak, B.; Dzienny, P.; Pozniak, B.; Tikhomirov, M.; Hoppe, V.; Ziolkowska, P.; Tylus, W.; Grzymajlo, M.; Gajadhur, A.; Antonczak, A. Cytotoxicity Study of UV-Laser-Irradiated PLLA Surfaces Subjected to Bio-ceramisation: A New Way towards Implant Surface Modification. Int. J. Mol. Sci. 2021, 22: 8436.
[R1] Sun, X.; Xue, B.; Yang, S. D.; Huo, K. W.; Liao, X. Y. Structural conversion of PLLA/ZnO composites facilitated by interfacial crystallization to potential application in oil-water separation. Appl. Surf. Sci. 2020, 517, 14613.
[R2] Sun, X.; Xue, B.; Tian, Y. Z.; Xie, L. 3D Porous Poly (L-lactic Acid) Materials with Controllable Multi-scale Microstructures and Their Potential Application in Oil-Water Separation. Appl. Surf. Sci. 2018, 4332, 32256-32261.
[R3] Guo, Y. F. , Sun, X., Xue, B., Zhou, Y., Xie, L. Carbon quantum dots-driven surface morphology transformation towards superhydrophobic poly(lactic acid) film. Colloid Surface A. 2023, 656, 130547.
[R4] Zheng, J. W., Yang , J. C., Cao, W., Huang, Y. Fabrication of transparent wear-resistant superhydrophobic SiO2 film via phase separation and chemical vapor deposition methods. Ceram Int. 2022, 48, 32143–32151.
[R5] Su, Y. Z.; Zhao, Y. Q.; Zheng, W. G. Asymmetric Sc-PLA Membrane with Multi-scale Microstructures: Wettability, Antifouling, and Oil−Water Separation. ACS Appl. Mater. Inter. 2020, 12, 55520−55526.
Q7: XRD patterns of carbon are amorphous and very broad. Please carefully evaluate the patterns with hkl tables.
A7: Thank you for your valuable suggestions which is of significance to improve our manuscript and work.
Fly ash (FA) mainly come from the flue gas emitted by burning coal. The main components of FA are SiO2, Al2O3, Fe2O3, and other metal oxides. A small amount of unburned carbon exists in FA, thus, the characteristic peaks of amorphous carbon are not observed in XRD pattern of FA. The wrong sentence has been corrected into “It was easy to find that FA exhibited the diffraction peak located at 26.7°, corresponding to the crystal phase quartz.”. In addition, the hkl parameters of FA are exhibited in Table. S1.
Table S1. XRD data of FA
|
Component |
2θ |
hkl |
|
SiO2 |
18.2° |
(312) |
|
SiO2 |
26.7° |
(513) |
|
Al2O3 |
35.4° |
(311) |
|
Fe2O3 |
33.3° |
(311) |
|
3Al2O3·2SiO2 |
40.1° |
(210) |
Q8: The reviewer misses tables with the data on roughness, composition, Tm/g, weight increase, delay time and CA (and roll-off angle) of the different surfaces. So the “pretty hydrophilicity” cannot be seen.
A8: Thank you for your valuable suggestions which is of significance to improve our manuscript and work. The data on composition, Tm, Tg, contact angle, and delay time are summarized in Table. S2. The “pretty hydrophilicity” has been revised into “unsatisfactory hydrophobicity”.
Table S2. Data analysis tables for pure PLLA and 5 wt% PLLA/FA@KH570 composite.
|
Sample Code |
Glass transition temperature (°C) |
Melting temperature (°C) |
Degree of crystallinity (%) |
Contact angles (°) |
Weight increase (%) |
Delay time (s) |
|
pure PLLA |
55.9 |
167.3 |
55.1 |
133 |
13.4 |
159 |
|
5 wt% PLLA/FA@KH570 |
58.9 |
166.7 |
78.2 |
151 |
5.3 |
410 |
Q9: Some images in the supporting information can improve the quality.
A9: Thank you for the professional comments. The supporting information has been added in the revised version.

Reviewer 2 Report
The authors proposed a superhydrophobic composite surface using fly ash and candle grease coating and its application to the anti-icing surface. The work is quite detailed and would be of interest to the scientific community. Please see below some aspects that can be improved.
1) How about stability of the surfaces in terms of the superhydrophobicity, especially the abrasion stability? This is very important for practical applications of the superhydrophobic surface.
2) Superhydrophobic surfaces are typically characterized by high contact angles of more than 150 degrees and contact angle hysteresis of less than 10 degrees (or very low sliding angles). So, the dynamic behavior of the droplets on the composite surface should also be mentioned.
Author Response
Response to the comments:
Dear Editors and Reviewers,
Thank you very much for your comments concerning our manuscript entitled “Integrating fly ash-controlled surface morphology and candle grease coating: access to highly hydrophobic poly (L-lactic acid) composite for anti-icing application” submitted to Nanomaterials. All the valuable comments have great help for our study and future work.
We have carefully read the comments and given detail answers as follows. We really appreciate and cherish this opportunity to answer these questions. A point-by-point revision to the reviewers’ criticisms has been made. Please find the final version with all corrections marked up using the “Track Changes”. We do hope our revised manuscript will meet your approval and be proceed further.
Sincerely yours,
Lan Xie
E-mail: mm.lanxie@gzu.edu.cn.
Response to Reviewer 2’s Comments:
Q1: How about stability of the surfaces in terms of the superhydrophobicity, especially the abrasion stability? This is very important for practical applications of the superhydrophobic surface.
A1: Thank you for your valuable suggestions which is of significance to improve our manuscript and work. The abrasion stability of PLLA/FA@KH570 composites is investigated in the revised manuscript.
Figure 7. (a) Contact angles of neat PLLA and PLLA/FA@KH570 composites. (b) Contact angles of neat PLLA and PLLA/FA@KH570 composites with candle grease coating. (c) Contact angles of neat PLLA and PLLA/FA@KH570 composites after abrasion.
As shown in Fig.7c, the apparent contact angle of PLLA/FA@KH570 composites after sandpaper abrasion was slightly decreased. For example, the contact angle of 5 wt% PLLA/FA@KH570 was moderately reduced from 151° to 145° after abrasion, which still exhibited the pretty hydrophobicity. The results indicated the excellent abrasion stability of PLLA/FA@KH570 composites.
Q2: Superhydrophobic surfaces are typically characterized by high contact angles of more than 150 degrees and contact angle hysteresis of less than 10 degrees (or very low sliding angles). So, the dynamic behavior of the droplets on the composite surface should also be mentioned.
A2: Thank you for the professional comments. The sliding angle measurement has been added into the revised.
As shown in Figure S5, the advancing and receding contact angles of 5 wt% PLLA/FA@KH570 at a tilt angle of 60° are 155.4° and 108.2°, respectively. Thus, the contact angle hysteresis up to 47.2° indicates the excellent adhesion of water droplet on the composite surface, in spite of the high static contact angle of 151°. According to the adhesion theory, rose petal-like surfaces were not only hydrophobic but also somewhat adherent. As could be seen from Figure 4, the surface of the 5 wt% PLLA/FA@KH570 composite had a petal-like structure, which is consistent with the Cassie impregnating wetting state. The more detailed information is seen in Supporting Information (Figure S2-S6). In addition, the “Superhydrophobic” has been corrected into “highly hydrophobic” in the revised manuscript.
Figure S5. Rolling angle photos of 5 wt% PLLA/FA@KH570 composite at the tilt angle of 30° (a) and 60° (b), respectively

Reviewer 3 Report
In the paper, the integration of morphology in a surface controlled by fly ash and with a grease coating for anti-icing coatings is studied. It is a very specific work without clear application to a specific group of materials. The application must be indicated in the text and collected in the conclusions.
It is well written but without describing the application there is no point in the investigation.
The bibliography should be enriched with non-Chinese citations. It is a very biased bibliography.
It must be corrected:
Line 191: Indicate which carbon phase is found. If there is quartz, indicate the peaks in which it is identified. All this can be pointed out in Figure 3.
In figure 4 the central photographs are not necessary. They must be deleted. A comment would suffice.
Line 259: what does DSC mean?
Figure 6: The footer must be explanatory. It must be changed.
Author Response
Response to the comments:
Dear Editors and Reviewers,
Thank you very much for your comments concerning our manuscript entitled “Integrating fly ash-controlled surface morphology and candle grease coating: access to highly hydrophobic poly (L-lactic acid) composite for anti-icing application” submitted to Nanomaterials. All the valuable comments have great help for our study and future work.
We have carefully read the comments and given detail answers as follows. We really appreciate and cherish this opportunity to answer these questions. A point-by-point revision to the reviewers’ criticisms has been made. Please find the final version with all corrections marked up using the “Track Changes”. We do hope our revised manuscript will meet your approval and be proceed further.
Sincerely yours, Lan Xie
E-mail: mm.lanxie@gzu.edu.cn.
Response to Reviewer 3’s Comments:
Q1: It is a very specific work without clear application to a specific group of materials. The application must be indicated in the text and collected in the conclusions. It is well written but without describing the application there is no point in the investigation.
A1: Thank you for your valuable suggestions which is of significance to improve our manuscript and work. The clear application of PLLA/FA@KH570 composites has been stated in the revised manuscript.
The FA@KH570-controlled surface morphology strategy had taken a significant step towards recycling waste FA and fabricating green hydrophobic PLLA composites, which had tremendous implication for anti-icing materials and even environment protection. The PLLA/FA@KH570 composite films had great potential as the promising refrigerator interior wall materials.
Q2: The bibliography should be enriched with non-Chinese citations. It is a very biased bibliography. It must be corrected.
A2: Thank you for your professional suggestions. Many Chinese citations in the bibliography has been changed into non-Chinese citations, including [3, 8-10, 13, 15, 36-38, 43, 46, 51-53, and 56].
Q3: Line 191: Indicate which carbon phase is found. If there is quartz, indicate the peaks in which it is identified. All this can be pointed out in Figure 3.
A3: Thank you for your valuable suggestions which is of significance to improve our manuscript and work.
It was easy to find that FA exhibited the diffraction peaks located at 18.2° and 26.7°, corresponding to the crystal phase quartz. Therein, the carbon content was low and the characteristic peaks were covered with quartz, which was imperceptible in the pattern.
Table S1. XRD data of FA
|
Component |
2θ |
hkl |
|
Quartz (SiO2) |
18.2° |
(312) |
|
Quartz SiO2 |
26.7° |
(513) |
|
Alumina (Al2O3) |
35.4° |
(311) |
|
Hematite (Fe2O3) |
33.3° |
(311) |
|
Mullite (3Al2O3·2SiO2) |
40.1° |
(210) |
Q4: In figure 4 the central photographs are not necessary. They must be deleted. A comment would suffice.
A4: Thank you for your valuable suggestions which is of significance to improve our manuscript and work. The central photographs in Figure 4 have been deleted, and the corresponding discussion has also been revised.
Figure 4. SEM images of (a-a1) pure PLLA, (b-b1) 1 wt% PLLA/FA@KH570, (c-c1) 3 wt% PLLA/FA@KH570, (d-d1) 5 wt% PLLA/FA@KH570, and (e-e1) 7 wt% PLLA/FA@KH570 composites.
Q5: Line 259: what does DSC mean?
A5: Thank you for the professional comments.
The full name of DSC is differential scanning calorimetry. The main application of DSC is to measure the thermal properties of the sample, including glass transition temperature, melting temperature, degree of crystallinity, and so on.
Q6: Figure 6: The footer must be explanatory. It must be changed.
A6: Sorry, it is our carelessness to make this mistake.
Figure 6. (a) DSC heating curves of pure PLLA and PLLA/FA@KH570 composites. (b) XRD patterns of pure PLLA and PLLA/FA@KH570 composites.

Round 2
Reviewer 1 Report
The text does not refer to any of the figures.
E.g. for roll test, refer to the figures in the supporting information.
you should mention in the conclusion that it is a rose petal surface (including a reference)
Author Response
Response to Reviewer 1’s Comments:
Q1: The text does not refer to any of the figures. E.g., for roll test, refer to the figures in the supporting information.
A1: Thank you for your valuable suggestions which is of significance to improve our manuscript and work. We have referred to Fig. S1-S7 and Table S1-S2 in the revised manuscript.
Q2: You should mention in the conclusion that it is a rose petal surface (including a reference).
A2: Thank you for your valuable suggestions which is of significance to improve our manuscript and work. It has been mentioned in the conclusion that PLLA/FA@KH570 composites exhibit the rose petal surface.
In addition, the contact angle hysteresis of 5 wt% PLLA/FA@KH570 composite was up to 47.2 °, indicating the excellent adhesion of water droplet on the rose petal surface.

Reviewer 3 Report
With the corrections introduced, the work has been better explained and easier to understand.
Author Response
Response to Reviewer 3’s Comments:
Q1: With the corrections introduced, the work has been better explained and easier to understand.
A1: Thank you for your valuable suggestions which is of significance to improve our manuscript and work. We are grateful for your positive comments on our manuscript.